# Factors Influencing the Droplet Size of Asphalt Emulsion during Fabrication

**Xiaowei Chen [1], Yan Meng [2], Guihua Hu [3], Ji Zhou [4] and Jian Ouyang [5],***

[1] Institute of Road and Bridge Engineering, Hunan Communication Engineering Polytechnic, Changsha 410132, China; chxwei910@163.com
[2] School of Civil Engineering, Dalian University of Technology, Dalian 116024, China; mengyan18@mail.dlut.edu.cn
[3] Quality and Safety Supervision Bureau of Transportation Construction in Hunan Province, Changsha 410116, China; huguihua20@163.com
[4] School of Civil and Environmental Engineering, Hunan University of Science and Engineering, Yongzhou 425199, China; hnkjxy_zhouji@163.com
[5] School of Transportation and Logistics, Dalian University of Technology, Dalian 116024, China
* Correspondence: ouyangjian@dlut.edu.cn or ouyangjian87@126.com

**Abstract:** The size distribution of asphalt droplets greatly affects the overall technical properties of asphalt emulsion, while it did not obtain much concern previously. In order to fabricate a good-quality asphalt emulsion with a small droplet size distribution, the effect of preparation parameters on the asphalt droplet size distribution, such as emulsifier dosage, asphalt temperature, shear time, pH value, and soap solution temperature, were systematically studied. All preparation parameters can affect the droplet size distribution of asphalt emulsion as well as the representative droplet diameters for the 10th, 50th and 90th cumulative volume percentile ($D_{10}$, $D_{50}$, and $D_{90}$). The order of preparation parameters are ranked as: emulsifier dosage > temperature of soap solution > pH of soap solution > shear time > asphalt temperature. Therefore, the emulsifier dosage, the temperature and pH of the soap solution should be carefully controlled to obtain asphalt emulsion with good droplet size distribution during fabrication. Compared to $D_{10}$, the $D_{50}$ and $D_{90}$ are more easily affected by the fluctuant preparation parameters, which are recommended to be utilized to evaluate the emulsifying effect of asphalt emulsion and judge the asphalt droplet size distribution.

**Keywords:** asphalt emulsion; droplet size distribution; emulsifier; pH value; temperature





## 1. Introduction

As traditional pavement materials, asphalt emulsion-based materials, which can be paved at ambient temperature, are widely used in pavement maintenance, preservation, and rehabilitation. Compared to hot asphalt materials, asphalt emulsion-based materials possess the merits of low energy consumption and carbon emission, thus they are more preferred in pavement engineering. Besides, their workability is less affected by temperature so they are more suitably used in thin asphalt layers than hot asphalt materials. Depending on the different application requirements, asphalt emulsion can be used as a binder for spraying materials (such as tack coat, prime coat, fog sealing layer, chip sealing) and mixing materials (e.g., slurry seal mixture, micro surfacing, cold recycled mixture and other technical forms) [1–6]. In the application, the performance of asphalt emulsion-based materials shows greater variability than that of hot asphalt materials. In this regard, how to reduce the performance variability of asphalt emulsion-based materials and guarantee their construction quality is a big issue in the application.

Generally, the performance variability of asphalt emulsion-based materials is highly related to the quality of asphalt emulsion. Asphalt emulsion is an emulsion in which asphalt droplets are dispersed in the emulsifier solution. The essential properties, i.e.,

the properties of residues, such as the asphalt droplet size distribution and the chemical stability of emulsifiers, can greatly affect the quality of asphalt emulsion. In the current specifications [7,8], many tests and indexes are used to evaluate the properties of asphalt emulsion, such as storage stability, adhesion performance, mixing stability and evaporation residue properties. These indices are mainly to evaluate the properties of residue and the chemical stability of the emulsion. The asphalt droplet size distribution does not obtain much concern in the current specifications. However, the asphalt droplet size distribution plays an important role in the fresh properties of the emulsion. According to Stoke's law, the sedimentation rate of a single droplet in the emulsion is highly related to the droplet's size and emulsion viscosity. Thus, the storage stability of asphalt emulsion has a good correlation with asphalt droplet size [9]. The smaller asphalt droplet sizes can be beneficial to the storage stability of asphalt emulsion. Meanwhile, the particle size distribution greatly affects the rheological properties of asphalt emulsion [10,11]. The decreasing mean asphalt droplet size can increase the viscosity of asphalt emulsion. The increase of viscosity restricts the movement of asphalt droplets, which can be also beneficial to the storage stability of asphalt emulsion. The droplet size also influences the residue on the sieve. The residue on the sieve decreases with the decrease of asphalt droplet size. In the aspect of emulsion-type prime coat, the small size of asphalt droplets is a precondition to ensure a good penetrative ability of asphalt emulsion in a densified base [2,12].

Expect for affecting the fresh properties of emulsion, the asphalt droplet size distribution can also affect the drying and demulsifying properties of emulsion, further affecting the performance of the asphalt emulsion-based mixture. In the studies of Ouyang et al. [13,14], asphalt emulsion with a smaller size distribution of asphalt droplets can have a more rapid drying behavior. Meanwhile, the demulsifying and film formation behavior of asphalt emulsion can be also improved with the decreasing mean size of asphalt droplets [13,14].

Overall, the size of asphalt droplets is an essential parameter that can greatly affect almost all properties of asphalt emulsion. It is believed that asphalt emulsion with a smaller size distribution of asphalt droplets can have better performance. However, because the droplet size of asphalt emulsion is not a mandatory index in the specification of asphalt emulsion, the asphalt droplet size distribution does not obtain much concern. As a result, how to obtain an asphalt emulsion with a smaller size distribution of asphalt droplets is still not very clear.

Based on the above consideration, the objective of this research is to know how to produce the asphalt emulsion with a smaller size distribution of asphalt droplets. To achieve this objective, factors influencing the droplet size of asphalt emulsion during fabrication are investigated. Generally, the emulsifying effect of an emulsifier on asphalt is related to the emulsifier dosage, pH value, temperature of asphalt and soap solution, and shear conditions [15–18]. The effect of these factors on the size distribution of asphalt droplets is studied. Besides, the factor sensitivity analysis is conducted, and then the key factors influencing the size distribution of asphalt droplets during fabrication are found. Overall, this work is beneficial to understanding how to fabricate a good-quality asphalt emulsion with small particle size distribution.

## 2. Materials and Experimental Methods

### 2.1. Asphalt Emulsions Preparation

A commercial cationic slow-setting emulsifier (coded as KZW) and basic asphalt with 60/80 penetration grade were used to fabricate asphalt emulsion. The main technical properties of asphalt are shown in Table 1. The emulsifier was produced by Tianjin Kangzewei Co. Ltd. in Tianjin, China. Hydrochloric acid was chosen to adjust the pH value of the soap solution. The effect of emulsifier dosage, shear time, pH value, and temperature of asphalt and soap solution on the size distribution of asphalt droplets was investigated, thus different emulsions were prepared, with preparation parameters listed in Table 2. The reasons for the levels of parametric variables in Tables 1 and 2 are as follows. According to the recommended dosage in the instruction manual of emulsifiers, emulsifier dosage

is from 3% to 5% in slow setting asphalt emulsion which can be used in cold mix asphalt. Cationic emulsifier is more active in acidic environments, so a pH value ranging from 1 to 5 was chosen. The asphalt temperature should ensure that the asphalt has a certain degree of fluidity, but it should not be very high to prevent soap from boiling. Therefore, the asphalt temperature was determined between 120 °C and 160 °C in this study. The temperature of soap solution also cannot be very high to prevent soap from boiling, thus the soap solution temperature was determined between 45 °C and 70 °C. The base asphalt with a penetration grade of 70 is normally used to fabricate asphalt emulsion, thus it is also used in this study. The shearing time is very important to the emulsifying effect of asphalt emulsion. However, long shearing times can generate heat that may affect the quality of asphalt emulsion. Therefore, the shear time between 0.5 min and 2 min is selected in this study. Asphalt emulsion is produced in a special colloid mill for asphalt emulsion, whose shear rate is fixed by the manufacturer. As can be seen in Table 2, the one-variate analysis method is used in this study. Only one factor is changed in a group. All emulsions have the basic formula in which the content of asphalt and soap solution (including water and emulsifier) is 60% and 40%, respectively. This formula is normally used in the real application of asphalt emulsion. A colloid mill with a capacity of 1 L was used to emulsify asphalt.

**Table 1.** Main technical properties of asphalt.

| Technical Property | Value |
|---|---|
| Softening point (°C) | 47.5 |
| Penetration at 25 °C (0.1 mm) | 70 |
| Ductility at 15 °C (cm) | >100 |
| Kinetic viscosity at 60 °C (Pa·s) | 227 |

**Table 2.** Preparation parameters of asphalt emulsions.

| Groups | Emulsifier Dosage (%) | Shear Time (min) | pH Value | Temperature (°C) | |
|---|---|---|---|---|---|
| | | | | Asphalt | Soap |
| 1 | 3, 4, 5 | 1 | 2 | 140 | 55 |
| 2 | 4 | 0.5, 1, 1.5, 2 | 2 | 140 | 55 |
| 3 | 4 | 1 | 1, 2, 3, 4, 5 | 140 | 55 |
| 4 | 4 | 1 | 2 | 120, 130, 140, 150, 160 | 55 |
| 5 | 4 | 1 | 2 | 140 | 45, 55, 65, 70 |

*2.2. Particle Size Distribution Test*

A laser particle size analyzer is shown in Figure 1 (LS-POP (9), Zhuhai OMEC Instruments Co. Ltd., Zhuhai, China) is used to test the size distribution of asphalt droplets. The equipment test range for particle size can be from 0.1 to 750 μm. Before the test, all asphalt emulsions were filtered by a filter screen of 1.18 mm to exclude large droplets. To obtain reasonable results, one or two drops of asphalt emulsion were firstly diluted with deionized water to ensure the shading rate of the specimen was between 10% and 20%. Then, the droplet size distribution of the specimen was automatically measured by the laser particle size analyzer (LS-POP (9), Zhuhai OMEC Instruments Co. Ltd., Zhuhai, China). Three representative droplet sizes ($D_{10}$, $D_{50}$, $D_{90}$) are selected for the analysis in this study. The $D_{10}$, $D_{50}$, and $D_{90}$ are the droplet diameter for the 10th, 50th and 90th cumulative volume percentile, respectively.

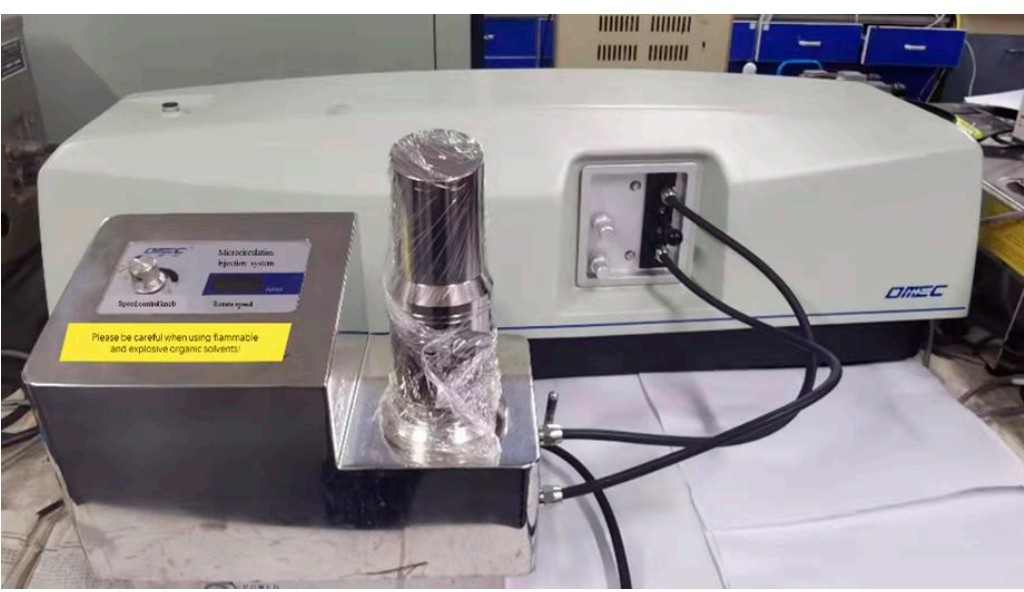

**Figure 1.** Laser particle size analyzer.

## 3. Results

### 3.1. Emulsifier Dosage

The effect of emulsifier dosage on the droplet size of asphalt emulsion is shown in Figure 2. It can be seen from Figure 2a that the droplet size of asphalt emulsions decreases with the increasing emulsifier dosages from 3% to 5%. Specifically, as shown in Figure 2b, the $D_{50}$ and $D_{90}$ of asphalt emulsion decreases quickly with the emulsifier dosage from 3% to 4% but then decreases slightly with the emulsifier dosage from 4% to 5%. The $D_{90}$ of asphalt emulsion with a 3% of emulsifier dosage is much larger than that of the other two emulsions. Therefore, 3% of the emulsifier dosage is not enough to produce the asphalt emulsion with smaller asphalt droplets. The function of an emulsifier is to reduce the interfacial tension between asphalt and water for emulsifying asphalt. It is reasonable that the emulsifying ability of soap solution can be greatly enhanced with the increasing emulsifier dosage. In order to ensure the emulsifying effect of asphalt emulsion, the employed emulsifier dosage should be no less than 4%. Besides, it should be noted that the $D_{10}$ differs little for the three asphalt emulsions, the reason will be discussed in the following.

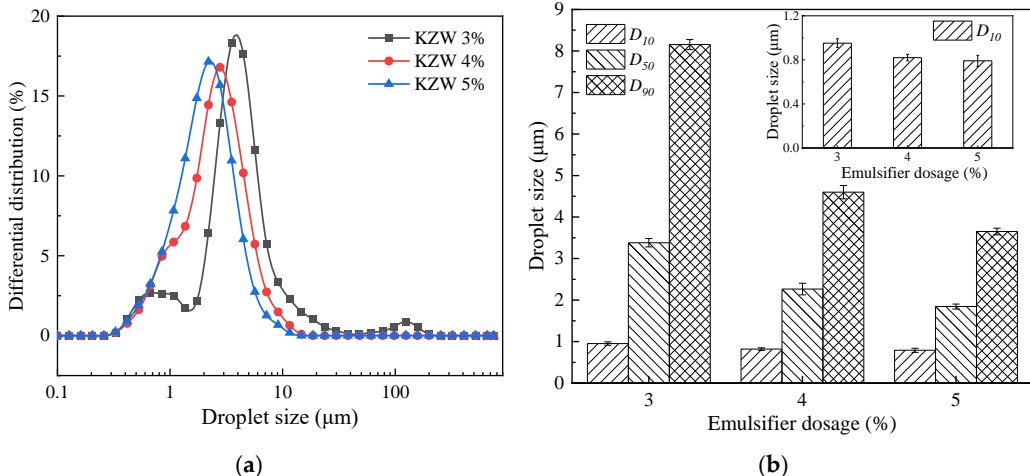

**Figure 2.** Droplet size distribution of asphalt emulsion with different emulsifier dosages. (**a**) Droplet size distribution; (**b**) Representative droplet sizes.

### 3.2. pH Value

The droplet size distribution of asphalt emulsions fabricated at different pH values is shown in Figure 3. It can be seen from Figure 3a that pH value can greatly affect the droplet size distribution of asphalt emulsions. As shown in Figure 3b, the $D_{50}$ and $D_{90}$ of asphalt emulsion firstly decreases but then increases with the increasing pH value. Asphalt emulsion with a pH value of 2 has the smallest representative droplet size. The asphalt emulsions with pH values of 4 and 5 have very larger representative droplet size compared to other emulsions. Generally, pH value can greatly affect the activity of the emulsifier. For cationic emulsifiers, emulsifier molecules can combine with hydrogen ions in the acidic condition, which is beneficial to the molecular activity and the emulsifying ability of the emulsifier. Because the emulsifying ability of the emulsifier can be improved with decreasing the pH value, the asphalt droplet size distribution is decreased accordingly. However, too low a pH value may affect the double electric layer structure of asphalt particles, which weakens the mutual repulsive force between asphalt droplets. Therefore, a pH value of 2 is recommended in the fabrication of asphalt emulsion.

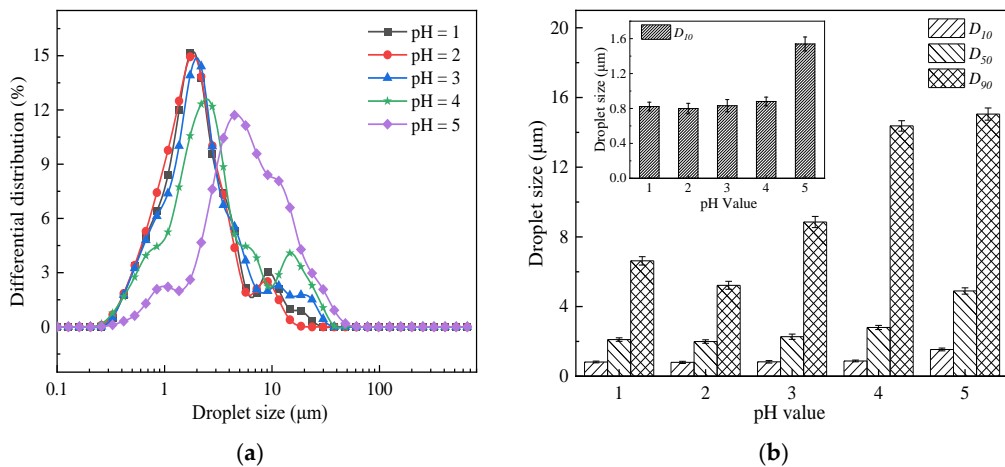

(a)  (b)

**Figure 3.** Droplet size distribution of asphalt emulsions fabricated at different pH values. (**a**) Drop let size distribution. (**b**) Representative droplet sizes.

It should be noted that the $D_{10}$ differs little for emulsions at different pH values except for that with a pH value of 5. Similar to emulsifier dosage, the $D_{10}$ is also insensitive to pH value.

### 3.3. Shear Time

Shear time refers to the emulsification time after the matrix asphalt and soap solution are poured into the colloid mill. The effect of shear time on the droplet size of asphalt emulsion is shown in Figure 4. It can be seen from Figure 4a that the shear time can moderately affect the droplet size distribution of asphalt emulsion. As shown in Figure 2b, the $D_{50}$ and $D_{90}$ of asphalt emulsion firstly decreases and then increases with the increasing shear time. The smallest $D_{50}$ and $D_{90}$ occurred at the shear time of 1.5 min and 1.0 min, respectively. Therefore, there is the optimum shear time for the emulsifying effect of asphalt emulsion. If the shear time is too short, the asphalt is not fully milled into smaller particles. However, when the shear time exceeds the optimum value, the excessive shear effect may lead to the coalescence of the droplets. Besides, the excessive high-shear effect can increase the temperature of asphalt emulsion, which can also lead to the droplets coalescing during cooling. Since the difference between $D_{90}$ between 1 min and 1.5 min is much larger than that of $D_{50}$ between 1 min and 1.5 min, the shear time with the lowest $D_{90}$ (1 min) is chosen as the optimum shear time.

### 3.4. Asphalt Temperature

The effect of asphalt temperature on the droplet size of the asphalt emulsion is shown in Figure 5. It can be seen from Figure 5a that asphalt emulsions fabricated at different asphalt temperatures show little difference in droplet size distribution. As shown in Figure 5b, the $D_{10}$ and $D_{50}$ of the five emulsions differ very little. Only the $D_{90}$ of the asphalt emulsion prepared at 130 °C is slightly larger than other emulsions. Therefore, the asphalt temperature ranging from 120 to 160 °C has little effect on the droplet size distribution of the asphalt emulsion.

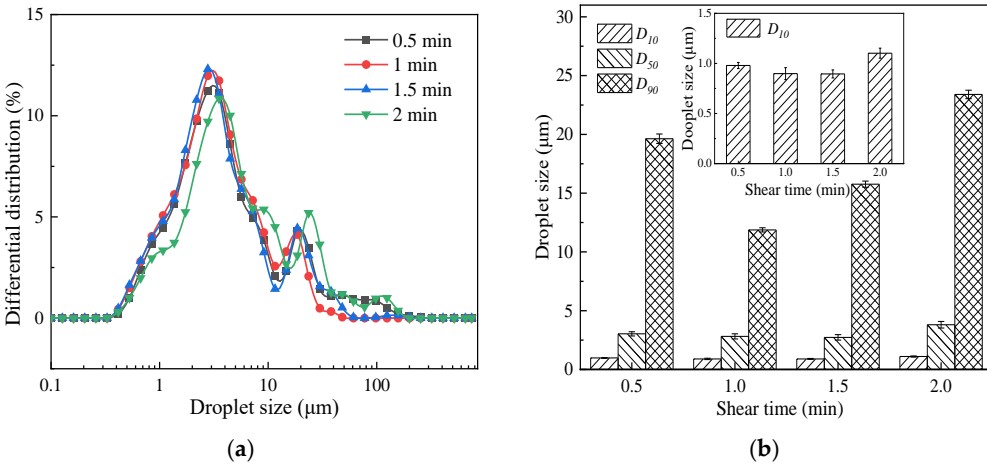

(**a**)　　　　　　　　　　　　　　　　　　　(**b**)

**Figure 4.** Droplet size distribution of asphalt emulsions fabricated at different shear time. (**a**) Droplet size distribution; (**b**) Representative droplet sizes.

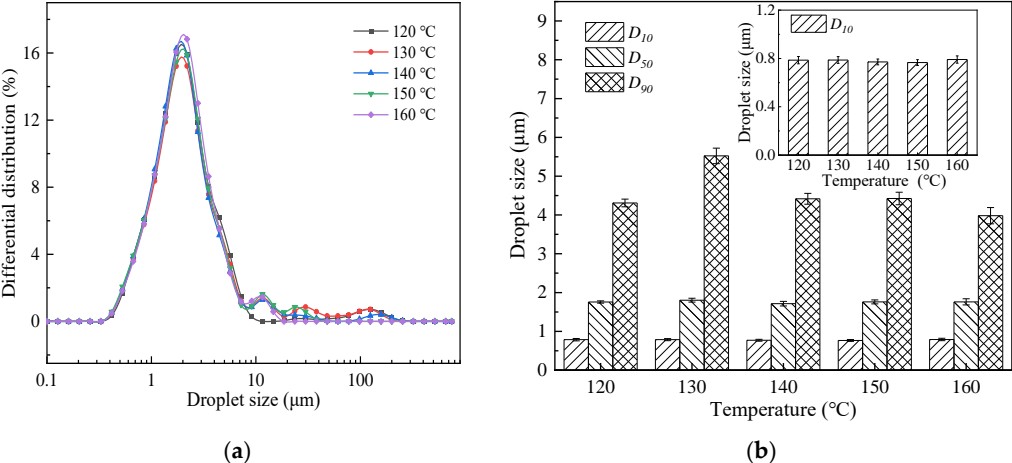

(**a**)　　　　　　　　　　　　　　　　　　　(**b**)

**Figure 5.** Droplet size distribution of asphalt emulsions fabricated at different asphalt temperatures. (**a**) Droplet size distribution; (**b**) Representative droplet sizes.

It should be stated here that the above results do not mean the asphalt temperature has little effect on the emulsifying effect of asphalt emulsion. The asphalt temperature can affect the residue content above the sieve of 1.18 mm, which is shown in Figure 6. It should be stated here that the residue content on the sieve is calculated as the ratio of the mass of asphalt on the sieve to the mass of the total emulsions. As can be seen from Figure 6, the residue content of asphalt emulsion on the sieve firstly decreases and then increases with the increase of asphalt temperature. Therefore, too low or too high asphalt temperature is not beneficial to fabricating the asphalt emulsion with very low residue content above the sieve of 1.18 mm. This phenomenon can be explained as follows. Because both the temperatures of asphalt and soap solution are not very high, the asphalt emulsion cannot be boiled during fabrication. In this asphalt temperature range, high temperature is beneficial

to the milling effect of asphalt and also the reactivity between asphalt and emulsifier. At low asphalt temperature, a small part of the asphalt may not be fully milled. However, although the asphalt can be well emulsified at high asphalt temperature, asphalt droplets can easily coalesce during cooling because of water evaporation, especially on the surface of asphalt emulsion. The skin phenomenon occurs due to evaporation during cooling [19]. To reduce this skin phenomenon during emulsion cooling, the asphalt temperature cannot be too high.

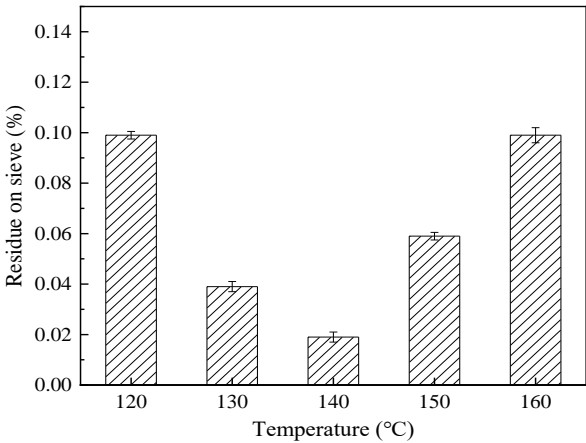

**Figure 6.** The residue content on the sieve of asphalt emulsion.

### 3.5. Temperature of Soap Solution

The effect of the temperature of soap solution on the droplet size of asphalt emulsion is shown in Figure 7. It can be seen from Figure 7 that the $D_{50}$ and $D_{90}$ of asphalt emulsion firstly decreases and then increases with the increasing soap temperature. The reason is similar to the effect of asphalt temperature. High temperature is beneficial to the molecular activity and emulsifying ability of the emulsifier. However, when the temperature of soap solution is very high, the temperature of asphalt emulsion is very high accordingly. In this condition, the phenomenon of droplets coalescence easily occurs during cooling around the surface of asphalt emulsion. Because of these two effects, the temperature of the soap solution should be moderate, around 55 °C for our study. Besides, the temperature of the soap solution has little effect on the $D_{10}$ for the four asphalt emulsions.

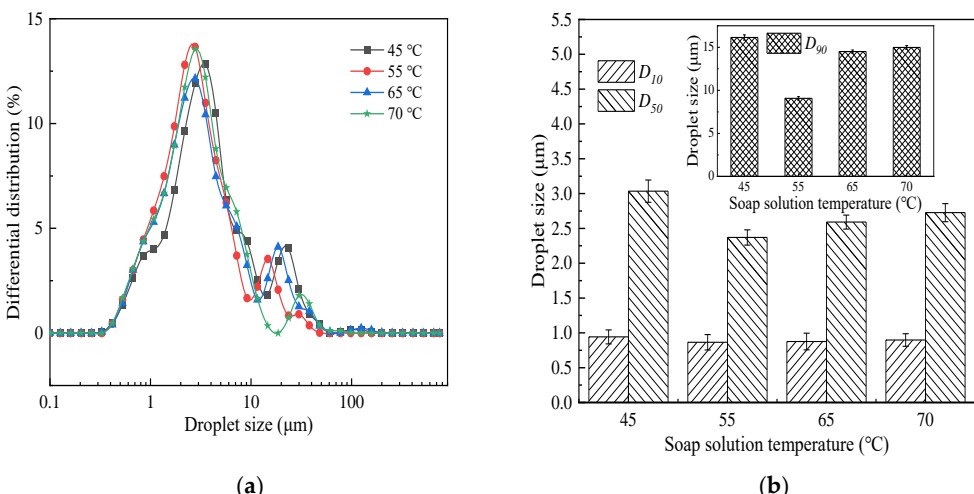

(**a**)          (**b**)

**Figure 7.** Effect of soap solution temperature on droplet size distribution of emulsion. (**a**) Droplet size distribution; (**b**) Representative droplet sizes.

### 3.6. Validation of Manufacturing Parameters

Based on the above consideration, it is recommended that the optimum manufacturing parameters are as follows: emulsifier dosage at 4%, pH value at 2, shear time at 1 min, asphalt temperature at 140 °C, the temperature of soap solution at 55 °C. To verify the effect of the optimum manufacturing parameters, an asphalt emulsion was fabricated and the droplet size distribution was tested. It can be seen from Figure 8 that the asphalt emulsion had a good droplet size distribution and the $D_{90}$ was smaller than 6 μm. Therefore, it is significant that the optimum manufacturing parameters are recommended in this study to fabricate a good quality asphalt emulsion.

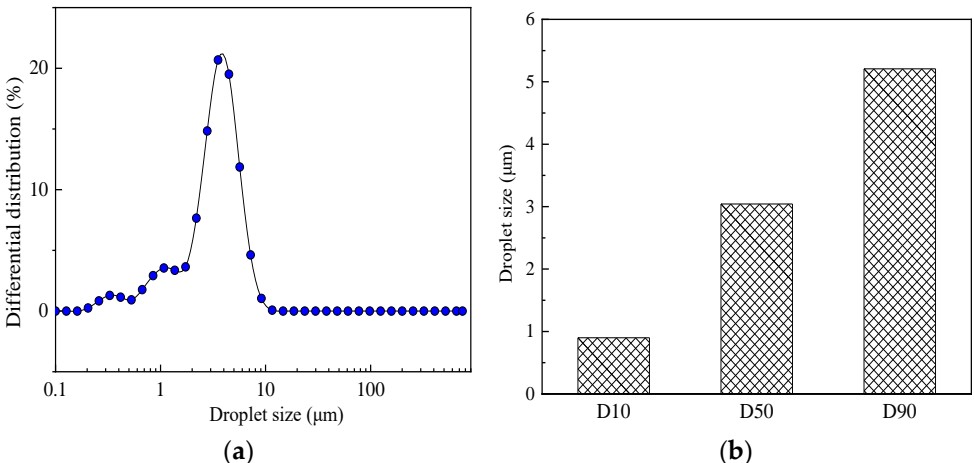

**Figure 8.** Droplet size distribution of asphalt emulsions fabricated at optimum manufacturing parameters. (**a**) Droplet size distribution. (**b**) Representative droplet sizes.

## 4. Discussion

### 4.1. Factors Sensitivity Analysis

According to the above results, the optimum preparation parameters of asphalt emulsion are obtained, which are listed in Table 3. In the real production of asphalt emulsion, these preparation parameters may be changed sometimes in the plant, which can affect the emulsifying effect of asphalt emulsion. Therefore, the effect of every factor on the representative droplet size of asphalt emulsion is studied. The sensitivity analysis ranges of every factor are shown in Table 3. The analysis points of the sensitivity are the two points around the optimum preparation parameter for every factor. For instance, because emulsifier dosage is 4%, emulsifier dosages of 3% and 5% are used to study the effect of emulsifier dosage on the sensitivity of the representative droplet size of asphalt emulsion.

**Table 3.** Optimum preparation parameters and their sensitivity analysis range of asphalt emulsion.

| Parameters | Value | Sensitivity Analysis Range |
|---|---|---|
| Emulsifier dosage (%) | 4 | 3–5 |
| pH value | 2 | 1–3 |
| Shear time (min) | 1 | 0.5–1.5 |
| Asphalt temperature (°C) | 140 | 130–150 |
| Soap solution temperature (°C) | 55 | 45–65 |

To visually and simply show the different effects of the above factors, the maximum change rate of the different representative droplet sizes are analyzed according to Equation (1). The representative droplet size at the optimum condition, and the representative droplet size of the two points before and after the optimum condition are chosen for analysis.

$$r_D = \frac{\max\left|D - D_{\text{optimum}}\right|}{D_{\text{optimum}}} \times 100\% \tag{1}$$

where $r_D$ is the maximum relative change of the representative droplet size. $D_{optimum}$ is the representative droplet size of asphalt emulsion under every factor with the optimum condition. $D$ is the representative droplet size of asphalt emulsion before and after the optimum condition.

The results of the effects of every factor on the representative droplet size of asphalt emulsion are shown in Figure 9 according to Equation (1). The value of $r_D$ indicates whether the factor is significant or not. It can be inferred from Figure 9 that the effect of every factor on the $D_{10}$ of asphalt emulsion is ranked as: emulsifier dosage > shear time > temperature of soap solution > pH of soap solution > asphalt temperature. The effect of every factor on the $D_{50}$ of asphalt emulsion is ranked as: emulsifier dosage > temperature of soap solution > pH of soap solution > shear time > asphalt temperature. The effect of every factor on the $D_{90}$ of asphalt emulsion is ranked as: temperature of soap solution > emulsifier dosage > pH of soap solution > shear time > asphalt temperature. Therefore, emulsifier dosage, temperature and pH of soap solution are main factors influencing the droplet size distribution of asphalt emulsion, and asphalt temperature has little effect on the droplet size distribution of asphalt emulsion. Therefore, in order to obtain asphalt emulsion with good droplet size distribution, the emulsifier dosage, the temperature and pH value of soap solution should be carefully controlled during the fabrication of asphalt emulsion. It should be stated here that although pH value can greatly affect the size distribution of asphalt droplets in Figure 3, the size distribution of asphalt droplets differs not too much when pH value is from 1 to 3. Thus, its effect on the asphalt droplet size is weaker than temperature of soap solution.

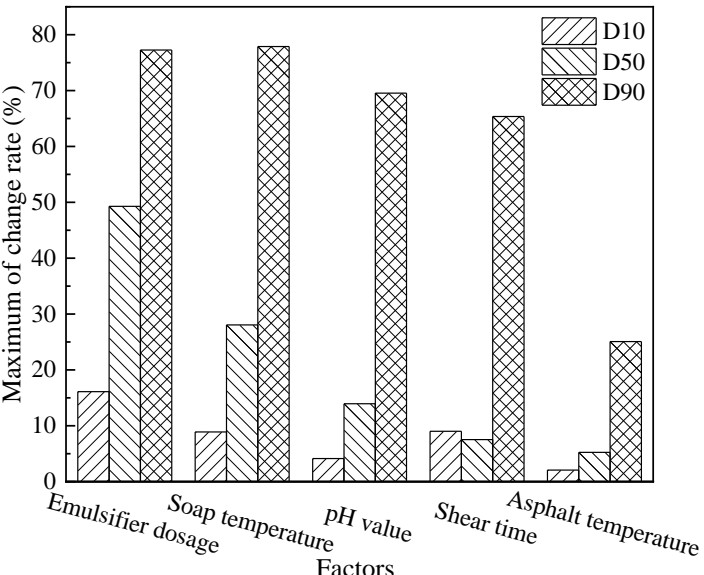

**Figure 9.** Factorial analysis about the droplet size of asphalt emulsion.

*4.2. Representative Droplet Size*

As mentioned in the section of introduction, asphalt droplet size distribution can greatly affect the technical properties of asphalt emulsion. Specifically, $D_{10}$ can greatly affect the rheological properties of asphalt emulsion because the non-Newtonian rheological behavior of the emulsion is highly related to the content of droplets with small size [20]. $D_{50}$ has a good correlation with the drying and film forming properties of asphalt emulsion [14,19]. $D_{90}$ is a representative droplet size for the content of droplets with lager size. Since large droplets can be easily settled in the emulsion, $D_{90}$ may greatly affect the storage stability of asphalt emulsion.

It can be seen from Figure 9 that the $D_{50}$ and $D_{90}$ are more easily affected by the fluctuant preparation parameters than the $D_{10}$ almost for every factor. Especially for $D_{90}$, the relative change can be higher than 60% with a slight wave in the preparation parameter.

Therefore, it is recommended to utilize the $D_{50}$ and $D_{90}$ to evaluate the emulsifying effect of asphalt emulsion and select the optimum asphalt emulsion as well as its optimum preparation parameters. The $D_{10}$ is affected less by the changing preparation parameters perhaps it is mainly dependent on the employed colloid mill.

## 5. Conclusions

The effect of emulsifier dosage, pH value, temperature of asphalt and soap solution, and shear time on the droplet size distribution of asphalt emulsion was studied. The representative droplet diameters for the 10th, 50th and 90th cumulative volume percentile ($D_{10}$, $D_{50}$, and $D_{90}$) were used in the discussion. On the basis of the research work discussed in this paper, the following conclusions can be drawn:

(1) All preparation parameters can affect the droplet size distribution of asphalt emulsion. Specifically, the representative asphalt droplet sizes ($D_{50}$ and $D_{90}$) are decreased with the increasing emulsifier dosage. The $D_{50}$ and $D_{90}$ are firstly decreased and then increased with the increasing pH value of soap solution, temperature of soap solution and shear time. The representative asphalt droplet sizes are little affected by asphalt temperature.

(2) Factor sensitivity analysis about the representative droplet diameters indicates that the order of factors influencing droplet size distribution is ranked as: emulsifier dosage > temperature of soap solution > pH of soap solution > shear time > asphalt temperature. Emulsifier dosage and temperature of soap solution are main factors influencing the asphalt droplet size distribution, especially for $D_{50}$.

(3) The $D_{50}$ and $D_{90}$ are more easily affected by the fluctuant preparation parameters than the $D_{10}$ almost for every factor. Therefore, it is recommended to utilize the $D_{50}$ and $D_{90}$ to evaluate the emulsifying effect of asphalt emulsion and judge the asphalt droplet size distribution.

**Author Contributions:** X.C.: methodology, investigation, writing—original draft. Y.M.: methodology, writing—original draft. G.H.: investigation, writing—original draft. J.Z.: investigation, writing—review and editing. J.O.: conceptualization, methodology, writing—review and editing. All authors reviewed the manuscript. All authors have read and agreed to the published version of the manuscript.

**Funding:** This research received no external funding.

**Institutional Review Board Statement:** Not applicable.

**Informed Consent Statement:** Not applicable.

**Data Availability Statement:** Not applicable.

**Acknowledgments:** The authors thank the Science Technology Innovation Project from Department of Transportation of Hunan province (202004) and the Natural Science Foundation of Hunan Province, China (Grant No. 2019JJ40093).

**Conflicts of Interest:** The authors declare that they have no conflict of interest.

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
