# Peer review of "Factors Influencing the Droplet Size of Asphalt Emulsion during Fabrication"

_coatings, doi:10.3390/coatings12050575_

Round 1
Reviewer 1 Report
The authors investigated the factors that affect the droplet size of asphalt emulsions. They conducted parametric laboratory work by selecting the emulsifier dosage, asphalt temperature, shear time, pH value, and soap solution temperature as asphalt emulsion manufacturing variables. They used the one-variate analysis method and all the other parameters were kept at a constant level while the studied variable changed in three to five levels. They used droplet diameters (D10, D50, and D90) to evaluate the effectiveness of the parameters and as a conclusion, the authors ranked the studied parameters as emulsifier dosage > temperature of soap solution > pH of soap solution > shear time > asphalt temperature.
My recommendations for the work can be found below.
- How are the levels of parametric variables selected? This requires a detailed explanation.
- According to my experience, one of the most important parameters for the droplet size of asphalt emulsions is the shear parameter. You studied just the duration of shear, what about the shear rate? Did you consider this effect also?
- The effects of droplet sizes on the engineering properties of emulsions need a more detailed discussion.
- It is indicated in the text that, before analyzing the particle size distribution of the samples, emulsions were filtered with a 1.18 mm filter to exclude large droplets. In section 3.4, where the effect of asphalt temperature on droplet size distribution was evaluated, the residue contents on the 1.18 mm sieve were given in figure 6. To my way of thinking it is important information and the residue percentages of all samples should be provided in the paper. In addition and just to be sure are the values presented in figure 6 ranging between 0.02 and 0.10% or 2 to 10%? And are these percentages calculated considering the total emulsion or just asphalt weight?
- Did you produce an asphalt emulsion using the optimum manufacturing parameters to check your results or not?
Reviewer 2 Report
Review on the manuscript entitled “Factors influencing the droplet size of asphalt emulsion during fabrication” by Xiaowei Chen et al.
The subject of the manuscript is of great interest considering the amount of asphalt used every year. The concern regarding a better understanding of size distribution of asphalt droplets is of interest due to continuous improvement in terms of quality. The purpose of this study consist in tests that investigated the effect of emulsifier dosage, pH value, shear time, temperature and soap solution on the asphalt droplets.
The manuscript is well written and explains clearly the results.
The design- figures and diagrams are explicit, readable, and of high resolution.
All the used materials and equipment are adequately described, with all the characteristics. A laser particle size analyzer was used for the accurate establishment of the size distribution of asphalt droplets.
The results provided by the experimental work done in this manuscript provides interesting information regarding the use of various factor’s influence on droplet diameter of asphalt and the optimum preparation parameters of asphalt emulsion.
The conclusions are well formulated and underline the findings of this research.
I kindly ask the authors to add more comparison with the literature data in the discussion part, therefore underling the findings that this research brings. Undoubtedly, this will add value to the research.
